# Effect of packaging in preventing cholesterol autoxidation in milk chocolates for a higher quality and safer shelf-life

Federico Canzoneri[1]☺*, Valerio Leoni[2]☺, Davide Risso[1]☺, Matteo Arveda[1]‡, Rosanna Zivoli[1]‡, Andrea Peraino[1]‡, Giuseppe Poli[3]☺, Roberto Menta[1]☺

**1** Soremartec Italia Srl, Ferrero Group, Alba, CN, Italy, **2** Laboratory of Clinical Chemistry, Hospital Pio XI of Desio, ASST-Brianza and School of Medicine and Surgery, University of Milano Bicocca, Monza, Italy, **3** Department of Clinical and Biological Sciences, San Luigi Hospital, University of Torino, Orbassano, TO, Italy

☺ These authors contributed equally to this work.
‡ MA, RZ and AP also contributed equally to this work.
* federico.canzoneri@ferrero.com

**Data Availability Statement:** All relevant data are within the paper.

**Funding:** Soremartec Italia Srl, Alba (CN), Italy. The financial support was provided in the form of

## Abstract

Non-enzymatic cholesterol oxidation products (COPs) are nowadays receiving increasing attention in food technology for their potential use as biomarkers of freshness and safety in raw materials and complex food matrices, as well as markers of cholesterol oxidation during the production and shelf-life of end products. Here reported is the investigation of how long three prototype milk chocolates containing whole milk powders (WMPs) of increasing shelf-lives (i.e. 20, 120, and 180 days), could be safely stored in the market by adopting the non-enzymatic COPs as a quality markers. In addition, the protective effect of two different primary packaging, sealed and unsealed ones, in mitigating the generation of non-enzymatic COPs in three prototype milk chocolates after 3, 6, 9, 12 months of shelf-life was assessed to simulate two real storage conditions. Quantifying oxysterols' levels by mass spectrometry, the oxygen impermeable packaging (PLUS) resulted to significantly quench the non-enzymatic COPs production up to 34% as to that found in the same product but with unsealed standard packaging (STD). This study represents one practical application of non-enzymatic COPs as a reliable tool for corrective strategies to prevent food oxidation.

## Introduction

Cholesterol oxidation products (COPs), mainly represented by oxysterols, have enzymatic or non-enzymatic origin. The oxysterols with an enzymatic origin showed immunomodulant, antiviral and metabolic activities and appear promising compounds in the pharmaceutical and biomedical fields [1], in particular 27-hydroxycholesterol (27OHC) [2]. The oxysterols of non-enzymatic origin more abundant in foodstuffs are 7β-hydroxycholesterol (7βOHC), 7-keto-cholesterol (7KC), 5α,6α-epoxycholesterol (α-epoxy), 5β,6β-epoxycholesterol (β-epoxy), and cholestan-3β,5α,6β-triol (triol): they result from cholesterol oxidation along food production and storage. Production and storage do also increase by autoxidation the levels of 7α-

salaries for authors Federico Canzoneri, Matteo Arveda, Rosanna Zivoli, Andrea Peraino and Roberto Menta who are employed by Soremartec Italia Srl, Alba (CN, Italy). At the time of conceptualization, Davide Risso was employed by Soremartec Italia Srl, while his current employer is Tate & Lyle Italy SpA. Giuseppe Poli and Valerio Leoni have scientific consultancy contracts with Soremartec Italia Srl.

**Competing interests:** Federico Canzoneri, Matteo Arveda, Rosanna Zivoli, Andrea Peraino and Roberto Menta are employed by Soremartec Italia Srl, Alba (CN, Italy). At the time of conceptualization, Davide Risso was employed by Soremartec Italia Srl, while his current employer is Tate & Lyle Italy SpA. Giuseppe Poli and Valerio Leoni have scientific consultancy contracts with Soremartec Italia Srl. The above mentioned authors were involved in the study design, collection, analysis, interpretation of data, the writing of this article and the decision to submit it for publication. All authors declare no other competing interests. This does not alter our adherence to PLOS ONE policies on sharing data and materials.

hydroxycholesterol (7αOHC) and 25-hydroxycholesterol (25OHC), two oxysterols that could be formed enzymatically as well [1], obviously not in food industry. The presence and the further formation of non-enzymatic oxysterols in food is unavoidable and there are several factors that can cause a net increase of them, in particular the manufacturing processes (handling process, heat process such as spray drying, frying, radiation), and the storage procedures (as exposure to oxygen, light, packaging conditions) [3].

Non-enzymatic COPs receive particularly growing attention in food technology for their potential use as biomarker of freshness in raw material and complex food matrices [4, 5] as well as markers of cholesterol oxidation along the manufacturing and the shelf-life of the final products.

Moreover, they are biologically active molecules involved in inflammation, apoptosis, cellular damage [6, 7] and in the pathogenesis of several degenerative diseases [8], thus for their importance on health and on the overall nutritional value of foods. Regarding the implications on health, these compounds may be altogether considered harmful, especially when they are introduced in excess, since they may contribute to the pathogenesis of metabolic disorders and sustain chronic inflammatory processes. Concerning the overall nutritional value of foods, these compounds can be considered as nutritional quality markers of foods [2]. 7βOHC and 7KC have been investigated as biomarker of the oxidation phenomena occurring during the production and storage of foodstuffs [3, 4, 9–15], hence they could also be employed as markers of corrective actions against the autoxidation of food components [2]. Moreover, several studies reported on the presence and possible accumulation of 7αOHC in foodstuffs [4, 15–20], but this oxysterol does not appear to have been adopted yet as a nutritional quality marker.

In a previous study, we verified the significance of COPs as biomarkers of cholesterol autoxidation and milk freshness in three prototype chocolates containing whole milk powders of increasing shelf-lives (20, 120, and 180 days). The increase of total oxysterols of non-enzymatic origin resulted to be proportional to the milk powder shelf-life, being 7βOHC and 7KC by far the most abundant cholesterol oxides [21], while the oxysterol of fully enzymatic origin, namely 27OHC did not show any significant variation as to the different freshness of the whole milk powder ingredient [21].

The here reported research was thus performed to investigate how long the three prototype milk chocolates could be safely kept on the market, adopting as a quality marker the generation of non-enzymatic cholesterol oxides, expanding the spectrum of non-enzymatic COPs analysis by including the quantification of 7αOHC as well. At the same time, the protective effect of two different food packaging in mitigating the generation of non-enzymatic COPs in the three prototype milk chocolates after 3, 6, 9 12 months of shelf-life was estimated (Fig 1).

Indeed, a suitable packaging appears to represent an affordable and very promising strategy of autoxidation reactions prevention in foodstuffs in general and in milk chocolate in the present case.

## Materials and methods

### Prototype milk chocolate production

As previously described by Risso et al. 2022 [21] three prototypes of milk chocolate tablets were produced according to a master recipe which included the following ingredients, in descending order of weight, sugar, 23.4% of whole milk powder (WMP), cocoa butter, cocoa liquor, soy lecithin and vanillin. The shelf-life of WMP in the recipe was considered as the key variable. Each milk chocolate sample was prepared by mixing chocolate liquor, sugar, WMP and cocoa butter in a mixer at 50˚C (Hobart FEG, Ohio, US) for a short time. The resulting mix was refined in a 3-roller refiner (Bühler Group, Uzwil, Switzerland) and then conched for

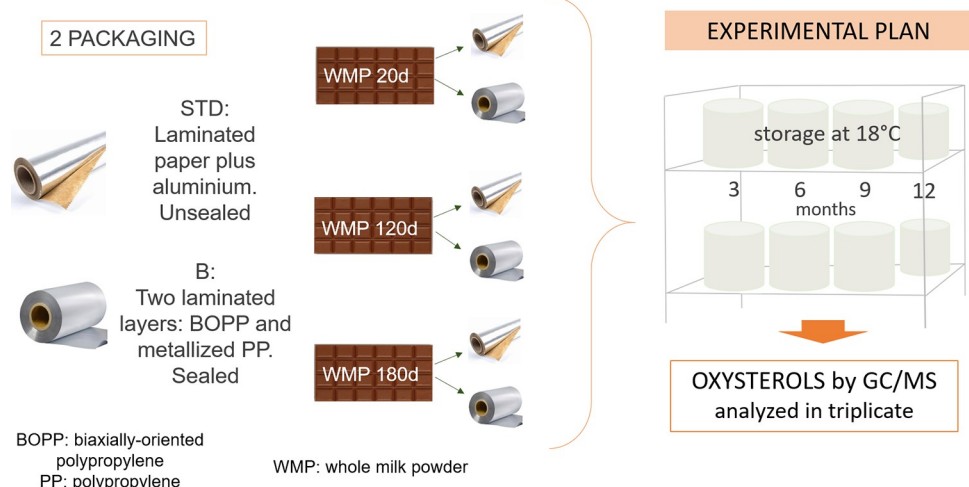

**Fig 1. Experimental research design.**

3 hours, at 50°C for the first 90 min and then at 45°C until 180 min. In the final conching step, vanillin and soy lecithin were added. All milk chocolate samples were manually tempered on a cold granite countertop and molded in plastic molds. After cooling, the molds were demolded, packed in plastic wrap, and stored in dark condition in a warehouse at 18°C until analysis. The analyses were conducted on different time points, at 0, 3, 6, 9, 12 months. The three milk chocolate samples were produced on the same day by the same operator following the same protocol to minimize external variables that might influence the results. Milk chocolate A contained fresh WMP with a shelf-life of 20 days. Milk chocolates B and C contained the same percentage of WMP from the same supplier but having a shelf-life of 120 and 180 days, respectively. The 12 months monitoring was included in the study design because this particular foodstuff is very shelf-life stable and the standard shelf-life recommendations for milk chocolate, sealed and properly maintained in the dark, for the purpose of best before date, go from 12 to 16 months, in tropical or temperate conditions, respectively [22].

## Packaging characteristics

Several suitable materials can be used in industry to package high-quality food such as, aluminum, paper, polyethylene terephthalate (PET), polystyrene (PS), polypropylene (PP), oriented polypropylene (OPP), that can be used alone or laminated, sealed or just folded around the product [23]. The two different packaging described below, were chosen as ideal candidates for the purpose of this study, as they simulate real storage conditions and, at the same time, they were comparing on a high protection scenario with a low protection one, the letter being used as standard reference.

The standard package (STD) was a composite packaging obtained by coupling with a solvent-based adhesive a layer of paper and a layer of flexible aluminum. The barrier characteristics of this packaging in term of water vapor transmission rate (WTR), oxygen transmission rate (OTR), volatile organic compounds (VOC), light, were in principle absolute, however, since the package was not hermetically sealed but only folded over the product, the barrier provided by STD packaging was only against the light, the choice of aluminum was led by mechanical characteristics only for the purpose of fold retention. The second package (PLUS) was made by two layers laminated by solvent-based adhesive: a layer of biaxially-oriented polypropylene (BOPP) and a layer of metallized PP with high optical density, providing very high

moisture and high oxygen barrier (very low WVTR and OTR). In addition, the pack was designed to be hermetically sealed and the metallized PP retains a high sealing integrity that ensured complete barrier against oxygen.

## Quantification of oxysterols and cholesterol

As previously described, oxysterols were quantified by gas chromatography-isotope dilution mass spectrometry [5, 21]. In brief, to a screw-capped vial sealed with a teflon septum, 500 mg of homogenized chocolate were added together with 50 μg of epicoprostanol (Sigma) and 50 ng of deuterium labeled oxysterols (d7-7βOHC, d7-7KC, d7-α-epoxy, d7-β-epoxy, d7-triol, d6-25OHC, d6-27OHC, all purchased by Avanti Polar Lipids) as internal standards and 50 μL of butylated hydroxytoluene (BHT, 5 g/L) and 50 μL of K3-EDTA (10 g/L) to prevent auto-oxidation.

Each vial was flushed with argon for 10 min to remove air. Alkaline hydrolysis was allowed to proceed at 4˚C overnight with magnetic stirring in the presence of ethanolic 2 M potassium hydroxide solution. After hydrolysis, the sterols were extracted with cyclohexane: 5 ml of cyclohexane were added to each vial, vortexed for 30s and allow to separate at 4˚C for 15 minutes. Finally, the vials were centrifuged at 3500 x g for 10 min at 4˚C. The upper cyclohexane phase was collected in a clean tube. This step was repeated twice. Thus, 3 mL of the cyclohexane extract were used for cholesterol analysis. The oxysterols were separated from cholesterol and sterols by elution of the remaining 7 mL on SPE cartridge (SI 100 mg columns, Isolute). The cyclohexane extract was dried under argon stream at room temperature. The residue was dissolved in 1 mL of toluene. Silicate cartridges were equilibrated with 3 × 1.0 mL n-hexane. The toluene dissolved samples were loaded onto a silica cartridge and allowed to pass through the column by gravity force at room temperature. The columns were washed with 8 × 1.0 mL 0.5% 2-propanol in n-hexane (v/v) followed by elution of the oxysterols with 5 × 1.0 mL 30.0% 2-propanol in n-hexane (v/v) as previously described. The organic solvents were evaporated under a gentle stream of argon and converted into trimethylsilyl ethers with 100 μL of BSTFA (70˚C for 60 min).

Analysis was performed by isotope dilution gas chromatography mass spectrometry (GC-MS) with aB-XLB column (30 m x 0.25 mm i.d. x 0.25 μm film thickness, J&W Scientific Alltech, Folsom, CA, USA.) in a HP 6890 NNetwork GC system (Agilent Technologies, USA) connected with a direct capillary inlet system to a quadrupole mass selective detector HP5975B inert MSD (Agilent Technologies, USA). GC system was equipped with a HP 7687 series autosamplers and HP 7683 series injectors (Agilent Technologies, USA).

The oven temperature program was as follows: initial temperature of 180˚C was held for 1 min, followed by a linear ramp of 20˚C/min to 270˚C, and then a linear ramp of 5˚C/min to 290˚C, which was held for 11 min. Helium was used as carrier gas at a flow rate of 1 mL/min and 1 μL of sample was injected in splitless mode. Injection was carried out at 250˚C with a flow rate of 20 mL/min. Transfer line temperature was 290˚C. Filament temperature was set at 150˚C and quadrupole temperature at 220˚C according with the manufacturer indication. Mass spectrometric data were acquired in selected-ion monitoring mode. Peak integration was performed manually, and oxysterols were quantified from selected-ion monitoring analysis against internal standards using standard curves for the listed sterols [5, 21].

Cholesterol was measured by GC-MS as previously published [5, 21], using the same GC program as described above. The 3 mL of the cyclohexane extract for cholesterol analysis were evaporated under argon and converted into trimethylsilyl ethers with 100 μL of BSTFA (70˚C for 60 min). Mass spectrometric data were acquired in selected-ion monitoring mode at m/z = 370 for epicoprostanol (internal standard) and m/z = 368 for cholesterol.

## Statistical analysis

Results are presented as mean ± SD.

The increment of COPs was calculated as the ratio between time point mean value and the t0 mean value for each preparation (A, B or C) both for standard and impermeable packaging. The % reduction of COPs formation was calculated as 100- (impermeable/standard packaging x 100) at any corresponding time point.

Significance of the observed reduction of cholesterol or increments of COPs was evaluated with ANOVA on repeated measures with Holm-Sidack method. The differences in COPs and cholesterol concentrations at any time point between standard and impermeable packaging were evaluated with Student's t-test (Sigmastat 4.0, Systat Software Inc., USA). Significance was set for $p < 0.05$.

# Results

## COPs in prototype milk chocolates at time 0

The cholesterol content of the three milk chocolate samples at time 0 was close in chocolates A (226.97 ± 2.84 ng/g) and B (223.13 ± 3.21 ng/g), and slightly but not significantly lower in chocolate C (199.84 ± 1.95 ng/g), the one made with whole milk with the longest shelf life (Tables 1–3). Such values were all actually similar to those reported in international food data system for milk chocolates [24] and in our previous prefatory study [21].

In agreement with our preceding study, a significant difference in the basal (time 0) content of non-enzymatic COPs was observed in the three prototype chocolate bars. The total non-enzymatic COPs, namely 7KC, 7αOHC, 7βOHC, α-epoxy, β-epoxy, triol, 25OHC, increased from 368.84 ± 10.24 ng/g in milk chocolate A (made with WMP of 20 days old) to 456.31 ± 17.41 ng/g (1.2-fold increase) in chocolate B (made with WMP of 120 days old) and 651.3 ± 20.98 ng/g (1.8-fold increase) in chocolate C ($p < 0.001$, Tables 1–3). Such a rise in bars B and C was clearly due to the longer shelf-life, and was supporting a relatively lower freshness of the milk powder employed as to bars A, made with fresh WMP.

Similarly, the non-enzymatic COPs/cholesterol ratio increased from 1.63 ± 0.04 ng/μg in chocolate A, to 2.05 ± 0.07 ng/μg in chocolate B (1.3-fold increase) and 3.26 ± 0.1 ng/μg in chocolate C (2-fold increase) ($p < 0.001$, Tables 1–3).

The main non-enzymatic COPs in the three chocolates were 7αOHC, 7βOHC and 7KC (86–87% of total non-enzymatic COPs, Tables 1–3).

## Non-enzymatic oxysterols recovered at the different shelf-lives in prototype milk chocolate bars packed with either standard aluminum foil (STD) or with a protective barrier against O$_2$ (PLUS)

In the milk chocolate A in STD package, the amount of total non-enzymatic oxysterols was 563.74 ± 19.97 ng/g, 1268.02 ± 24.77 ng/g, 1417.87 ± 36.79 ng/g and 1673.33 ± 39.28 ng/g respectively after 3, 6, 9 and 12 months of storage, the latter mimicking a placing on the market ($p < 0.001$). The percent increase of non-enzymatic oxysterols as to time 0 resulted to be of 1.5, 3.4, 3.8 and 4.5-fold after 3, 6, 9 and 12 months of storage, respectively, at any time point significantly higher as to the preceding one ($p < 0.001$, Table 1). Conversely, in the milk chocolate A in PLUS package, in comparison to the same bars in STD package, the increased of total non-enzymatic oxysterols resulted to be significantly quenched. In fact, as reported in Table 1, the percent increase of non-enzymatic oxysterols as to time 0 was in this case of 1.3, 2.7, 3.0 and 3.4-fold after 3, 6, 9 and 12 months of storage ($p < 0.001$), respectively. Since the COPs amount was significantly reduced ($p < 0.001$) at any time point, in the case of the PLUS

**Table 1. Cholesterol (μg/g) and COPs[1] (ng/g) in prototype milk chocolate bars made with WMP of 20 days of shelf-life (Milk Chocolate A), packed with laminated paper and aluminum, unsealed (STD) or with two laminated layers of biaxially-oriented polypropylene (BOPP) and metallized polypropylene (PP), sealed (PLUS), then kept on virtual market for 0, 3, 6, 9, 12 months.** Values are expressed as means ± SD (n = 3). At any time point, the levels of non-enzymatic COPs in the Milk Chocolate A in packaging PLUS resulted to be significantly lower than those estimated in the Milk Chocolate A in STD packaging (§ $p < 0.001$).

| Samples | Cholesterol | 7α-hydroxycholesterol (7αOHC) | 7β-hydroxycholesterol (7βOHC) | 7-ketocholesterol (7KC) | 5,6α-epoxycholestanol (α-epoxy) | 5,6β-epoxycholestanol (β-epoxy) | cholestan-3β,5α,6β-triol (triol) | 25-hydroxycholesterol (25OHC) | Total non-enzymatic COPs | Non-enzymatic COPs/cholesterol ratio (ng/μg) | 27-hydroxycholesterol (27OHC) |
|---|---|---|---|---|---|---|---|---|---|---|---|
| Milk Chocolate A-T0 | 226.97 ± 2.84 | 108.01 ± 3.55 | 127.14 ± 1.09 | 81.04 ± 3.43 | 13.39 ± 0.94 | 18.9 ± 0.57 | 1.39 ± 0.08 | 18.97 ± 0.58 | 368.84 ± 10.24 | 1.63 ± 0.04 | 93.38 ± 3.02 |
| Milk Chocolate A-T3-STD | 192.58 ± 2.97 | 145.2 ± 5.6 | 188.78 ± 5.15 | 139.71 ± 6.52 | 23.25 ± 0.85 | 32.99 ± 0.54 | 2.42 ± 0.23 | 31.39 ± 1.07 | 563.74 ± 19.97 | 2.93 ± 0.08 | 83.2 ± 2.69 |
| Milk Chocolate A-T3-PLUS | 202.96 ± 3.75 | 125.38 ± 6.08 | 151.21 ± 4.86 | 128.51 ± 2.79 | 17.15 ± 0.74 | 21.15 ± 0.39 | 1.77 ± 0.08 | 27.51 ± 0.6 | 472.69 ± 15.53§ | 2.33 ± 0.05 | 87.67 ± 3.52 |
| Milk Chocolate A-T6-STD | 182.19 ± 3.4 | 180.79 ± 4.02 | 239.51 ± 3.83 | 209.05 ± 5.05 | 179.25 ± 5.13 | 376.01 ± 3.06 | 13.37 ± 0.39 | 70.04 ± 3.29 | 1268.02 ± 24.77 | 6.96 ± 0.01 | 69.37 ± 2.68 |
| Milk Chocolate A-T6-PLUS | 187.13 ± 5.58 | 156.08 ± 2.78 | 208.89 ± 6.01 | 207.14 ± 7.77 | 142.07 ± 5.35 | 234.85 ± 6.69 | 7.51 ± 0.19 | 53.44 ± 3.46 | 1009.98 ± 32.26§ | 5.4 ± 0.02 | 72.19 ± 0.61 |
| Milk Chocolate A-T9-STD | 169.23 ± 4.36 | 203.41 ± 6.89 | 263.2 ± 5.5 | 248.87 ± 3.47 | 210.2 ± 9.36 | 396.58 ± 8.95 | 15 ± 0.73 | 80.61 ± 1.88 | 1417.87 ± 36.79 | 8.38 ± 0.01 | 64.87 ± 2.75 |
| Milk Chocolate A-T9-PLUS | 178.29 ± 2.11 | 172.04 ± 4.09 | 227.25 ± 5.32 | 230.53 ± 6.48 | 166.17 ± 6.34 | 251.32 ± 5.68 | 9.35 ± 0.27 | 61.75 ± 2.35 | 1118.41 ± 30.53§ | 6.27 ± 0.14 | 70.01 ± 2.58 |
| Milk Chocolate A-T12-STD | 157.20 ± 3.95 | 225.47 ± 5.68 | 309.42 ± 6.95 | 322.29 ± 7.63 | 246.09 ± 5.54 | 461.91 ± 9.61 | 19.04 ± 0.71 | 89.1 ± 3.16 | 1673.33 ± 39.28 | 10.64 ± 0.02 | 61.32 ± 2.02 |
| Milk Chocolate A-T12-PLUS | 176.49 ± 3.23 | 191.74 ± 5.63 | 269.91 ± 4.02 | 254.93 ± 6.86 | 190.65 ± 5.23 | 277.38 ± 6.55 | 10.19 ± 0.45 | 68.82 ± 1.51 | 1263.62 ± 30.25§ | 7.16 ± 0.06 | 64.7 ± 2.15 |

[1] COPs: cholesterol oxidation products.

**Table 2. Cholesterol (µg/g) and COPs[1] (ng/g) in prototype milk chocolate bars made with WMP of 120 days of shelf-life (Milk Chocolate B), packed with laminated paper and aluminum, unsealed (STD) or with two laminated layers of biaxially-oriented polypropylene (BOPP) and metallized polypropylene (PP), sealed (PLUS), then kept on virtual market for 0, 3, 6, 9, 12 months.** Values are expressed as means ± SD (n = 3). At any time point, the levels of non-enzymatic COPs in the Milk Chocolate B resulted to be significantly higher than those measured in the Milk Chocolate A (p < 0.001). On the contrary, at any time point, the levels of non-enzymatic COPs in the Milk Chocolate B in packaging PLUS resulted to be significantly lower than those estimated in the Milk Chocolate B in STD packaging (§ p < 0.001).

| Samples | Cholesterol | 7α-hydroxycholesterol (7αOHC) | 7β-hydroxycholesterol (7βOHC) | 7-ketocholesterol (7KC) | 5,6α-epoxycholestanol (α-epoxy) | 5,6β-epoxycholestanol (β-epoxy) | cholestan-3β,5α,6β-triol (triol) | 25-hydroxycholesterol (25OHC) | Total non-enzymatic COPs[1] | Non-enzymatic COPs[1]/cholesterol ratio (ng/µg) | 27-hydroxycholesterol (27OHC) |
|---|---|---|---|---|---|---|---|---|---|---|---|
| Milk Chocolate B-T0-STD | 223.13 ± 3.21 | 124.74 ± 5.35 | 158.94 ± 4.62 | 107.73 ± 5.07 | 18.52 ± 1.01 | 24.99 ± 0.93 | 2.24 ± 0.07 | 19.15 ± 0.36 | 456.31 ± 17.41 | 2.05 ± 0.07 | 83.08 ± 3.23 |
| Milk Chocolate B-T3-STD | 187.76 ± 1.73 | 169.6 ± 5.16 | 220.74 ± 4.59 | 199.12 ± 3.73 | 34.29 ± 0.84 | 42.32 ± 1.6 | 3.75 ± 0.12 | 35.33 ± 1.71 | 705.14 ± 17.74 | 3.76 ± 0.08 | 77.72 ± 2.59 |
| Milk Chocolate B-T3-PLUS | 198.27 ± 2.65 | 153.16 ± 3.13 | 190.49 ± 5.88 | 159.07 ± 5.88 | 24.78 ± 0.93 | 33.62 ± 0.49 | 2.93 ± 0.13 | 29.48 ± 0.94 | 593.54 ± 17.38§ | 2.99 ± 0.07 | 81.49 ± 0.53 |
| Milk Chocolate B-T6-STD | 172.2 ± 4.88 | 199.33 ± 8.39 | 258.72 ± 4.59 | 266.65 ± 3.61 | 264.48 ± 7.24 | 399.23 ± 12.64 | 13.19 ± 0.9 | 71.545 ± 1.97 | 1473.15 ± 39.34 | 8.55 ± 0.02 | 61.85 ± 3.65 |
| Milk Chocolate B-T6-PLUS | 192.94 ± 2.26 | 174.39 ± 3.01 | 225.98 ± 2.92 | 252.58 ± 3.48 | 206.76 ± 4.88 | 323.04 ± 7.25 | 11.72 ± 0.58 | 64.13 ± 2.32 | 1258.6 ± 24.43§ | 6.52 ± 0.07 | 70.84 ± 2.69 |
| Milk Chocolate B-T9-STD | 166.17 ± 3.28 | 220.31 ± 6.02 | 285.66 ± 5.84 | 291.16 ± 7.08 | 299.88 ± 8.35 | 433.65 ± 5.07 | 17.07 ± 1.08 | 81.85 ± 2.72 | 1629.58 ± 36.16 | 9.81 ± 0.03 | 57.28 ± 1.43 |
| Milk Chocolate B-T9-PLUS | 186.99 ± 2.73 | 203.39 ± 4.84 | 241.86 ± 5.39 | 276.85 ± 7.29 | 231.71 ± 6.57 | 349.83 ± 6.69 | 14.71 ± 0.86 | 72.17 ± 2.79 | 1390.52 ± 34.45§ | 7.44 ± 0.11 | 67.77 ± 1.89 |
| Milk Chocolate B-T12-STD | 159.81 ± 3.27 | 262.66 ± 10.23 | 356.62 ± 8.71 | 351.4 ± 9.14 | 342.73 ± 7.38 | 487.55 ± 14.46 | 19.44 ± 0.88 | 88.23 ± 3.99 | 1908.63 ± 54.79 | 11.94 ± 0.14 | 54.35 ± 2.72 |
| Milk Chocolate B-T12-PLUS | 177.06 ± 2.98 | 239.4 ± 9.85 | 281.16 ± 5.87 | 310.97 ± 3.78 | 254.78 ± 5 | 371.36 ± 13.01 | 15.84 ± 0.78 | 82.79 ± 1.76 | 1556.30 ± 40.06§ | 8.78 ± 0.11 | 66.27 ± 2.24 |

[1] COPs: cholesterol oxidation products.

**Table 3. Cholesterol (µg/g) and COPs[1] (ng/g) in prototype milk chocolate bars made with WMP of 180 days of shelf-life (Milk Chocolate C), packed with laminated paper and aluminum, unsealed (STD) or with two laminated layers of biaxially-oriented polypropylene (BOPP) and metallized polypropylene (PP), sealed (PLUS), then kept on virtual market for 0, 3, 6, 9, 12 months.** Values are expressed as means ± SD (n = 3). At any time point, the levels of non-enzymatic COPs in the Milk Chocolate C resulted to be significantly higher than those measured in the Milk Chocolate B ($p < 0.001$). On the contrary, at any time point, the levels of non-enzymatic COPs in the Milk Chocolate C in packaging PLUS resulted to be significantly lower than those estimated in the Milk Chocolate C in STD packaging (§ $p < 0.001$).

| Samples | Cholesterol | 7α-hydroxycholesterol (7αOHC) | 7β-hydroxycholesterol (7βOHC) | 7-ketocholesterol (7KC) | 5,6α-epoxycholestanol (α-epoxy) | 5,6β-epoxycholestanol (β-epoxy) | cholestan-3β,5α,6β-triol (triol) | 25-hydroxycholesterol (25OHC) | Total non-enzymatic COPs[1] | Non-enzymatic COPs[1]/cholesterol ratio (ng/µg) | 27-hydroxycholesterol (27OHC) |
|---|---|---|---|---|---|---|---|---|---|---|---|
| **Milk Chocolate C-T0-STD** | 199.84 ± 1.95 | 203.29 ± 6.17 | 218.6 ± 3.03 | 142.77 ± 8.71 | 26.36 ± 0.92 | 31.14 ± 1.08 | 5.44 ± 0.27 | 23.7 ± 0.8 | 651.3 ± 20.98 | 3.26 ± 0.1 | 70.01 ± 1.34 |
| **Milk Chocolate C-T3-STD** | 190.37 ± 1.27 | 264.47 ± 6.05 | 309.18 ± 11.99 | 283.28 ± 5.88 | 47.11 ± 1.29 | 47.06 ± 1.14 | 8.32 ± 0.41 | 41.75 ± 2.17 | 1001.18 ± 28.92 | 5.26 ± 0.17 | 62.03 ± 0.71 |
| **Milk Chocolate C-T3-PLUS** | 195.92 ± 1.92 | 203.69 ± 4.39 | 254.04 ± 4.98 | 211.56 ± 5.13 | 30.43 ± 1.19 | 34.34 ± 1.01 | 6.85 ± 0.24 | 28.71 ± 1.59 | 769.61 ± 18.52§ | 3.93 ± 0.08 | 67.76 ± 1.7 |
| **Milk Chocolate C-T6-STD** | 179.36 ± 3.64 | 310.14 ± 6.58 | 401.14 ± 8.91 | 435.06 ± 7.86 | 388.23 ± 12.12 | 482.17 ± 11.81 | 35.42 ± 2.06 | 74 ± 2.22 | 2126.17 ± 51.57 | 11.85 ± 0.07 | 52.32 ± 2.09 |
| **Milk Chocolate C-T6-PLUS** | 185.61 ± 5.2 | 236.62 ± 2.9 | 316.16 ± 2.49 | 299.36 ± 9.78 | 221.13 ± 5.37 | 338.48 ± 5.5 | 24.18 ± 0.92 | 55.86 ± 1.86 | 1491.8 ± 28.82§ | 8.04 ± 0.1 | 55.09 ± 1.55 |
| **Milk Chocolate C-T9-STD** | 172.34 ± 4.32 | 356.37 ± 8.68 | 496.29 ± 9.85 | 493.73 ± 13.35 | 441.73 ± 13.11 | 552.39 ± 9.86 | 41.68 ± 1.43 | 82.45 ± 2.2 | 2464.65 ± 58.49 | 14.3 ± 0.03 | 49.02 ± 1.9 |
| **Milk Chocolate C-T9-PLUS** | 183.61 ± 1.45 | 263.39 ± 6.75 | 341.7 ± 10.07 | 325.84 ± 7.28 | 250.72 ± 6.74 | 363.43 ± 5.52 | 28.93 ± 0.82 | 61.41 ± 2.58 | 1635.42 ± 39.75§ | 8.91 ± 0.21 | 50.85 ± 1.51 |
| **Milk Chocolate C-T12-STD** | 161.35 ± 2.85 | 401.36 ± 8.68 | 553.97 ± 10.32 | 560.16 ± 6.99 | 524.78 ± 11.5 | 617.72 ± 14.58 | 44.95 ± 1.11 | 89.42 ± 2.37 | 2792.36 ± 55.56 | 17.31 ± 0.06 | 44.48 ± 1.68 |
| **Milk Chocolate C-T12-PLUS** | 168.38 ± 4.41 | 290.28 ± 7.65 | 395.39 ± 11.34 | 391.59 ± 13.87 | 307.83 ± 14.04 | 406.76 ± 7.58 | 31.42 ± 0.84 | 67.3 ± 2.21 | 1890.58 ± 57.53§ | 11.23 ± 0.07 | 47.96 ± 1.24 |

[1] COPs: cholesterol oxidation products.

packaging, the "quenching oxysterol power" of the impermeable packaging to oxygen was about 21%.

In the milk chocolate B, in the STD package, total non-enzymatic oxysterols, already 1.2-fold higher than in bars A at time zero (Table 2), kept increasing at 3, 6, 9 and 12 storage time ($p < 0.001$), showing values 1.15–1.2-fold higher than those accumulated by chocolate A in STD package ($p < 0.001$), with the extension of the storage time. In fact, the percent increase of total non-enzymatic oxysterols as to time 0 resulted to be of 1.5, 3.2, 3.6 and 4.2-fold after 3, 6, 9 and 12 months of storage ($p < 0.001$), respectively (Table 2). Again, the PLUS packaging allowed to significantly quench ($p < 0.001$) the shelf-life dependent increase of non-enzymatic oxysterols in chocolate B bars. Indeed, when comparing the total amount of these cholesterol oxides quantified in PLUS versus STD chocolate B bars, at the different experimental points, a quenching oxysterol power of PLUS package of 16% ($p < 0.001$), 15% ($p < 0.001$), 15% ($p < 0.001$) and 18% ($p < 0.001$) after 3, 6, 9 and 12 months of storage, was observed (Table 2).

As for chocolate A, also in chocolate B the mean level of total non-enzymatic cholesterol detected after 12 months storage in the bars in PLUS package resulted to be very close to that found in the bars in STD package at 6 months of shelf-life.

In the milk chocolate C, in the STD package, total non-enzymatic oxysterols, already 1.8-fold higher than in bars A at time zero ($p < 0.001$, Table 3), showed a progressive increase as well with the extension of the storage time, with values at 3, 6, 9 and 12 of shelf-life, being respectively higher as to time 0 of 1.5, 3.3, 3.8 and 4.3-fold ($p < 0.001$, Table 3). The PLUS packaging showed in the case of chocolate C versus STD chocolate C bars, a quenching oxysterol power of PLUS package of 23% ($p < 0.001$), 30% ($p < 0.001$), 34% ($p < 0.001$) and 32% ($p < 0.001$) after 3, 6, 9 and 12 months of storage, practically identical to that observed in chocolate B PLUS versus chocolate B STD (Table 3).

Also, in the chocolate C prototype, the PLUS packing allowed to limit the increase of total non-enzymatic oxysterols with the extension of shelf-life, with the result to obtain in the chocolate bars stored 12 months a content very similar to that reached at 6 months of storage by identical chocolate bars but in STD package.

As regards 27OHC, this was the only oxysterol of unique enzymatic origin detectable in the investigated food products. The other major fully enzymatic oxysterol, namely 24-hydroxycholesterol, was not detectable in the cow milk used to make the chocolate prototypes (Leoni, Risso and Poli, unpublished observation). In chocolate A-STD, made with fresh whole milk powder, 27OHC was present in the same concentration range of 7αOHC, 7βOHC, 7KC (Table 1), while in chocolate B-STD and C-STD, made with much seasoned WMPs, its concentration as to that of the three non-enzymatic oxysterols dropped down from 0.5 up to 3-fold ($p < 0.001$, Tables 2 and 3). The PLUS packaging just slightly quenched the loss of 27OHC consistently observed in prototype chocolate bars at all experimental times (Tables 1–3).

## Main features of the progressive increase of non-enzymatic oxysterols in the three chocolate prototypes

The storage of the three chocolates implied a progressive increase with time of the autoxidation of the cholesterol ingredient, with a consequent progressive accumulation of oxysterols, clearly those of the non-enzymatic type only. This fact applies to 7αOHC and 25OHC as well, theoretically of both enzymatic and non-enzymatic origin, for the absence of the relative generating enzymes in the products tested.

Quantitative and qualitative difference were observed in the progressive accumulation of non-enzymatic oxysterols, in the three tested chocolates. From the quantitative point of view, the oxysterols accumulation at the different shelf-lives investigated was relatively moderate at 3

months of storage, and then resulting more than three times as higher as at time in all three chocolates in STD package. A quite similar, but quantitatively lower, trend was observed in all three PLUS packaged chocolate bars (Tables 1–3). The sharp increase in the rate of oxysterols accumulation shown in all investigated conditions starting from 6 months of shelf-life appears to be mainly due to an exhaustion of the antioxidants originally present in the different chocolate ingredients and especially in the WMPs. As an example, the tocopherols, that are quite represented in the whole milk, donate an electron to the oxidant species generated during the storage without any possibility to be reconstituted by receiving an electron back.

From the qualitative point of view, in all the three prototypes A, B and C, either packaged with aluminum foil or with the protective barrier against oxygen and analyzed over 12 months of shelf-life, the dominant oxysterols at month 3 were 7αOHC, 7βOHC and 7KC (84–86% and 85–87% of total non-enzymatic oxysterols in STD and PLUS bars, respectively), while α-epoxy and β-epoxy were detected in very low amount (9–11% and 8–10%, respectively in STD and PLUS bars) (Tables 1–3). Of interest, during further storage, starting from month 6, α-epoxy and β-epoxy increased to a greater extent and were detected in larger quantities (41–45% and 37–42% of total non-enzymatic oxysterols in STD and PLUS bars, respectively), at the expense of the 7αOHC, 7βOHC and 7KC levels (49–54% and 52–57%, respectively in STD and PLUS bars) (Tables 1–3). 25OHC and triol as well showed a sharper increase from shelf-life of 6 months onwards, but their quantitative contribution to non-enzymatic oxysterols accumulation was minor in all experimental conditions investigated (Tables 1–3). The net shift in the type of oxysterols generated observed with the prolongation of the chocolate bars storage in all conditions tested points to a likely modulation of the kinetics of cholesterol autoxidation, an event certainly worth of further analysis in another context, more physico-chemically oriented.

## Discussion

The present report provides a comprehensive picture of the generation of oxysterols due to the cholesterol oxidation when chocolate bars made with whole milk powder are stored for defined periods of time. Such a production is relatively moderate at short shelf-lives, then it sharply increases once the antioxidant defenses provided by the various chocolate ingredients, in particular the whole milk powder, are exhausted.

Clearly, the oxysterols that are produced during prolonged storage of cholesterol containing food products are exclusively of non-enzymatic origin. The only oxysterol that uniquely has an enzymatic origin among those detectable in food products, namely 27OHC, was quite well represented in the three chocolate prototypes before the storing procedure (time 0), but it showed a gradual quantitative decrease with the progression of the storage time, in all investigated conditions (Tables 1–3). Such a progressive loss of 27OHC is likely due to some yet unclear degradation process, but certainly not dependent upon enzymatic reactions. Such degradation process would partly imply oxygen addition, since the PLUS packaging, by providing a barrier against oxygen, consistently while slightly quenched it (Tables 1–3).

In the here reported study, the non-enzymatic oxysterols, recently re-proposed as useful markers of quality in cholesterol-rich food products like chocolate [1, 2, 21], have been analyzed in a very comprehensive way. Not only selected non-enzymatic oxysterols as often reported in the literature as occurring in a variety of foodstuffs rich in cholesterol, but all range of compounds of this class so far described in food.

Non-enzymatic oxysterols as candidate markers of food freshness. Indeed, in the chocolate prototype A made with 20 days old WMP, the basal level of these compounds resulted to be much lower than that found in chocolate prototype B (with 120 days old WMP), the highest

basal oxysterols' content being recovered in chocolate prototype C (containing 180 days old WMP) (Tables 1–3). Therefore, the extent of the increase of non-enzymatic oxysterols at the different shelf-lives investigated always resulted to be inversely related to the degree of WMP freshness, both in STD and PLUS packaged chocolate bars.

Non-enzymatic oxysterols appear to be also good markers of food safety. In fact, these cholesterol oxides were repeatedly and consistently shown to exert toxic and pro-inflammatory effects in various human tissues and organs [7, 8, 25]. Primarily exposed to non-enzymatic oxysterols present in the diet the intestinal epithelial layer. Besides the net pro-oxidant and cytotoxic effects that could be exerted on the gut epithelium, an oxysterol mixture compatible with that occurring in the gut of people fed a Western diet [26] appears to strongly up-regulate the expression and synthesis of pro-inflammatory cytokines in differentiated Caco2 human intestinal cells [27]. Furthermore, the same oxysterol mixture was more recently demonstrated to heavily impair levels and correct localization of the main components of the intestinal tight junctions, namely, zonula occludens-1 (ZO-1), occludin and adhesion molecule a (JAM-A) [28]. A similar result was achieved by challenging CaCo-2 differentiated cells with either 7βOHC or 7KC in the low micromolar range [29]. Even if 7βOHC and 7KC were consistently proven to be the most toxic oxysterols [7], also cholesterol epoxides, whose relative concentration markedly increased in all experimental conditions from six months of shelf-life onwards (Tables 1–3), may be considered potentially toxic. For instance, α-epoxy was shown to up-regulate the activity of NADPH oxidase-1 (NOX1) in differentiated CaCo-2 cells, and, through up-regulated NOX1, to induce apoptotic cell death [30]. Moreover, both α-epoxy and β-epoxy has been for a long time possibly involved in the multistep process of carcinogenesis [31]. The latter aspect being a very good reason no to extend too much the shelf-life of milk, in particular whole milk, containing food at least.

The derangement of the tight junctions of gut epithelium as possibly exerted by excessive amounts of dietary oxysterols would make the gut epithelium itself permeable to the gut microbiota, to opportunistic pathogens in particular, with consequent trigger of immune and inflammatory response, finally determining a dangerous impairment of the physiological gut-brain axis [25].

Hence, a suitable containment of the accumulation of non-enzymatic oxysterols during the production in this case of chocolate bars and even more during their placing on the market clearly represents a must to properly contribute to the product's safety and quality.

"Quality", because through processes of autoxidation possibly occurring in foodstuffs and reliably monitored by oxysterols' quantification, food ingredients and food products are depleted of important nutritional factors, like lipids, including cholesterol, antioxidants (e.g. tocopherols and polyphenols) and, last but not least, also proteins, that rapidly undergo oxidation so loosing proper structure and function [32].

For all these reasons, besides focusing on fresh ingredients in food industry, in this case in chocolate industry, improved and more efficient procedures to prevent or at least markedly quench the non-enzymatic reactions of nutrients with environmental oxidants must be pursued. With this in mind, a virtual placing on the market of three whole milk chocolate prototypes, made with fresh WMP (chocolate A) or 120 or 180 days of shelf-life WMPs (chocolate B and C, respectively) was designed and all non-enzymatic oxysterols so far reported in animal food ingredients or products have been monitored up to one year of shelf-life, comparing a standard packaging (STD) to an advanced one (PLUS), that guaranteed a full barrier to oxygen, besides that to the light, also provided by the STD package. Indeed, the PLUS packaging afforded a quite good quenching of cholesterol autoxidation. The best proof of this result has been the observation that the mean level of total non-enzymatic oxysterols detected after 12 months storage in bars A, B and C in PLUS package resulted to be similar to that found in bars

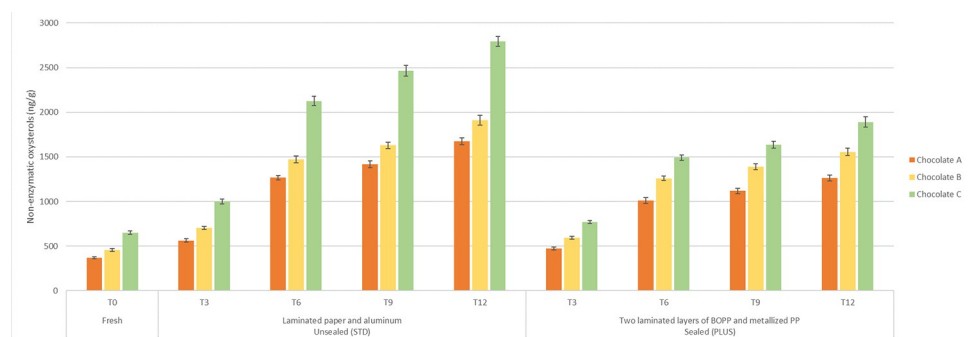

**Fig 2. Distribution of non-enzymatic cholesterol oxidation products (COPs) (ng/g) in prototype milk chocolate bars made with whole milk powder (WMP) of 20, 120 and 180 days of shelf-life (Milk Chocolate A, B and C, respectively), packed with laminated paper and aluminum, unsealed (STD) or with two laminated layers of BOPP and metallized polypropylene (PP), sealed (PLUS), kept on virtual market for 0, 3, 6, 9, 12 months.** Values are expressed as means ± SD (n = 3).

A, B and C in STD package already after 6 months of storage (Tables 1–3; Fig 2). This important finding was further strengthened by the COPs/cholesterol ratio values calculated at the relative time points (Tables 1–3).

## Conclusions

The inappropriate use of ingredients of animal origin (always containing cholesterol) in food and, even more, inappropriate storage conditions of food products may strongly enhance the production and accumulation of oxysterols, all of non-enzymatic origin. First, the quality of such foodstuff may be markedly affected, in terms of loss of important nutritional constituents, lipids, proteins, antioxidants. Second, but extremely important, the safety of such unproperly obtained and stored foodstuff could be likely affected.

The here reported results point out the mandatory need of choosing a suitable protective barrier for the prevention of a series of oxidative phenomena that can reduce the nutritional quality of a food product.

While nowadays it is common practice to sell milk chocolate up to 12 months of shelf-life, the various messages stemming from this study strikingly point to a more rigid limitation of the storage time, unless efficient protection against the autoxidation of biological ingredients is afforded.

To the best of our knowledge, this is the first report of quantitative and comprehensive monitoring of the non-enzymatic oxysterols detectable in milk chocolate during production and commercialization. More important, this study provided a likely useful comparison of the protective effect of a packaging essentially preventing photooxidation during the placing on the market as to that afforded by an advanced packaging in addition providing an efficient barrier to oxygen.

## Author Contributions

**Conceptualization:** Davide Risso, Roberto Menta.

**Data curation:** Valerio Leoni.

**Formal analysis:** Valerio Leoni.

**Investigation:** Federico Canzoneri, Davide Risso, Giuseppe Poli.

**Methodology:** Valerio Leoni, Matteo Arveda, Rosanna Zivoli, Andrea Peraino, Giuseppe Poli.

**Project administration:** Federico Canzoneri.

**Supervision:** Giuseppe Poli, Roberto Menta.

**Validation:** Valerio Leoni, Giuseppe Poli.

**Visualization:** Federico Canzoneri.

**Writing – original draft:** Federico Canzoneri, Valerio Leoni, Davide Risso, Matteo Arveda, Rosanna Zivoli, Andrea Peraino, Giuseppe Poli.

**Writing – review & editing:** Federico Canzoneri, Valerio Leoni, Davide Risso, Giuseppe Poli, Roberto Menta.

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
