## [Decision Letter · Decision Letter 0]

20 Feb 2023

PONE-D-22-28821Effect of packaging in preventing cholesterol autoxidation in milk chocolates for a higher quality and safer shelf-lifePLOS ONE

Dear Dr. Canzoneri,

Thank you for submitting your manuscript to PLOS ONE. After careful consideration, we feel that it has merit but does not fully meet PLOS ONE’s publication criteria as it currently stands. Therefore, we invite you to submit a revised version of the manuscript that addresses the points raised during the review process.

We look forward to receiving your revised manuscript.

Kind regards,

Brahma Nand Singh

Academic Editor

PLOS ONE

Journal Requirements:

This research was funded by Soremartec Italia Srl, Alba (CN), Italy.

Federico Canzoneri, Matteo Arveda, Rosanna Zivoli, Andrea Peraino and Roberto Menta are employed by Soremartec Italia Srl, Alba (CN, Italy). At the time of conceptualization, Davide Risso was employed by Soremartec Italia Srl, while his current employer is Tate & Lyle Italy SpA. Giuseppe Poli and Valerio Leoni have scientific consultancy contracts with Soremartec Italia Srl.

Federico Canzoneri, Matteo Arveda, Rosanna Zivoli, Andrea Peraino and Roberto Menta are employed by Soremartec Italia Srl, Alba (CN, Italy). At the time of conceptualization, Davide Risso was employed by Soremartec Italia Srl, while his current employer is Tate & Lyle Italy SpA. Giuseppe Poli and Valerio Leoni have scientific consultancy contracts with Soremartec Italia Srl. The above mentioned authors were involved in the study design, collection, analysis, interpretation of data, the writing of this article and the decision to submit it for publication. All authors declare no other competing interests.

We note that one or more of the authors are employed by a commercial company: name of commercial company. 

Reviewers' comments:

Reviewer's Responses to Questions

**Comments to the Author**

1. Is the manuscript technically sound, and do the data support the conclusions?

Reviewer #1: Yes

2. Has the statistical analysis been performed appropriately and rigorously? 

Reviewer #1: No

3. Have the authors made all data underlying the findings in their manuscript fully available?

Reviewer #1: Yes

4. Is the manuscript presented in an intelligible fashion and written in standard English?

Reviewer #1: Yes

5. Review Comments to the Author

Reviewer #1: The authors investigated how long the three prototype milk chocolates could be safely kept on the market, adopting as a quality marker the generation of non-enzymatic cholesterol oxides, expanding the spectrum of non-enzymatic COPs analysis by including the quantification of 7αOHC as well. At the same time, the protective effect of two different food packaging in mitigating the generation of non-enzymatic COPs in the three prototype milk chocolates after 3, 6, 9, 12 months of shelf-life was determined. The results demonstrate a suitable packaging represents an affordable and very promising strategy to prevent the autoxidation reactions in milk chocolate.

This study is well prepared and organized. The results are good and thoroughly discussed. Some major changes are required.

In the introduction: The authors should introduce the packaging materials that have been frequently used in packaging milk chocolate.

In materials and methods: Statistical analysis should be added for data analysis.

Packaging characteristics section should test the physical properties (such as WVP, OP, mechanical properties) of the two different packaging materials selected.

Data in the tables should be statistically analyzed.

6. PLOS authors have the option to publish the peer review history of their article (what does this mean?). If published, this will include your full peer review and any attached files.

Reviewer #1: No

---

## [Author Response · Author response to Decision Letter 0]

6 Mar 2023

Dear Reviewer 1, 

hereafter please find our point-by-point response to your questions, advises and comments. Thanks for the time you are spending in handling and also revising our manuscript.

1. In the introduction: The authors should introduce the packaging materials that have been frequently used in packaging milk chocolate.

This information is already present in the packaging characteristic section, we have now modified the abstract adding some information on the packaging materials used in the study.

2. In materials and methods: Statistical analysis should be added for data analysis.

We have now added a paragraph with the statistical analysis.

3. Packaging characteristics section should test the physical properties (such as WVP, OP, mechanical properties) of the two different packaging materials selected.

In the packaging characteristics section, we have now added some information about WTR, OTR, VOC.

4. Data in the tables should be statistically analyzed.

We have done the statistical analysis and now added the significance in the tables with related comments in the text.

---

## [Editor Report · Decision Letter 1]

6 Apr 2023

Effect of packaging in preventing cholesterol autoxidation in milk chocolates for a higher quality and safer shelf-life

PONE-D-22-28821R1

Dear Dr. Canzoneri,

We’re pleased to inform you that your manuscript has been judged scientifically suitable for publication and will be formally accepted for publication once it meets all outstanding technical requirements.

Kind regards,

Brahma Nand Singh

Academic Editor

PLOS ONE
---

## [Editor Report · Acceptance letter]

12 Apr 2023

PONE-D-22-28821R1 

Effect of packaging in preventing cholesterol autoxidation in milk chocolates for a higher quality and safer shelf-life 

Dear Dr. Canzoneri:

I'm pleased to inform you that your manuscript has been deemed suitable for publication in PLOS ONE. Congratulations! Your manuscript is now with our production department. 

Kind regards, 

on behalf of

Dr. Brahma Nand Singh 

Academic Editor

PLOS ONE